# Antibiotic Resistance Rates for *Helicobacter pylori* in Rural Arizona: A Molecular-Based Study

**DOI:** 10.3390/microorganisms11092290

**Published:** 2023-09-12

**Authors:** Fernando P. Monroy, Heidi E. Brown, Claudia M. Acevedo-Solis, Andres Rodriguez-Galaviz, Rishi Dholakia, Laura Pauli, Robin B. Harris

**Affiliations:** 1Department of Biological Sciences, College of the Environment, Forestry and Natural Sciences, Northern Arizona University, 617 South Beaver Street, Flagstaff, AZ 86011, USA; 2Pathogen and Microbiome Institute, Northern Arizona University, Flagstaff, AZ 86011, USA; 3Department of Epidemiology and Biostatistics, Mel and Enid Zuckerman College of Public Health, 1295 N Martin Ave, Tucson, AZ 85724, USArbharris@arizona.edu (R.B.H.); 4Winslow Indian Health Care Center, 500 North Indiana Avenue, Winslow, AZ 86047, USAlaura.pauli@wihcc.org (L.P.)

**Keywords:** *Helicobacter pylori*, antibiotic resistance, mutation rates, American Indian

## Abstract

*Helicobacter pylori* (*H. pylori*) is a common bacterial infection linked to gastric malignancies. While *H. pylori* infection and gastric cancer rates are decreasing, antibiotic resistance varies greatly by community. Little is known about resistance rates among rural Indigenous populations in the United States. From 2018 to 2021, 396 endoscopy patients were recruited from a Northern Arizona clinic, where community *H. pylori* prevalence is near 60%. Gastric biopsy samples positive for *H. pylori* (*n* = 67) were sequenced for clarithromycin- and metronidazole-associated mutations, 23S ribosomal RNA (23S), and oxygen-insensitive NADPH nitroreductase (*rdxA*) regions. Medical record data were extracted for endoscopic findings and prior *H. pylori* history. Data analysis was restricted to individuals with no history of *H. pylori* infection. Of 49 individuals, representing 64 samples which amplified in the 23S region, a clarithromycin-associated mutation was present in 38.8%, with T2182C being the most common mutation at 90%. While the prevalence of metronidazole-resistance-associated mutations was higher at 93.9%, the mutations were more variable, with D95N being the most common followed by L62V. No statistically significant sex differences were observed for either antibiotic. Given the risk of treatment failure with antibiotic resistance, there is a need to consider resistance profile during treatment selection. The resistance rates in this population of American Indian patients undergoing endoscopy are similar to other high-risk populations. This is concerning given the high *H. pylori* prevalence and low rates of resistance testing in clinical settings. The mutations reported are associated with antibiotic resistance, but clinical resistance must be confirmed.

## 1. Introduction

*Helicobacter pylori* (*H. pylori*) is among the most common bacterial infections in the world and one of the most common infectious agents linked to any malignancy (~89% of non-cardia gastric cancer [1]). Eradication of *H. pylori* is recommended to reduce the risk of gastric cancer and improve gastritis and gastric atrophy [2]. Testing for the infection is recommended for patients with a history of *H. pylori*-associated disease, and confirmation testing for eradication should be performed [2]. Unfortunately, culturing *H. pylori* for antibiotic-susceptibility testing is not routinely performed in Indian Health Service clinics.

Treatment for *H. pylori* typically includes a proton pump inhibitor and a combination of antibiotics which should be adjusted based on the resistance rates within the geographic region. Multiple factors affect treatment success, with compliance and antibiotic resistance as the two most commonly cited reasons for failure [3]. More than three-quarters of respondents in a national survey indicated compliance with eradication therapies was difficult due to gastrointestinal upset [4]. For this reason, patient support and treatment follow-up is important. Commonly used therapies are estimated to fail in approximately 25–40% of patients now [5,6,7], primarily due to increasing antibiotic resistance.

*H. pylori* antibiotic resistance rates vary greatly between each antibiotic and across geographic communities, and resistance rates are increasing [8]. A systematic review of antibiotic resistance in the Asia-Pacific regions found overall resistance for clarithromycin around 17% (95% CI 15–18) and 44% (95% CI 39–48) for metronidazole, with clarithromycin resistance rates increasing over the 1990–2016 time period of the study [9]. A review of articles published between 2007 and 2017 found clarithromycin rates in the Americas to be near 10% (95% CI 4–16%) with metronidazole rates of 23% (91% CI 2–44%) [10]. The observed resistance prevalence rates were lower when the studies were restricted to only include those studies among treatment-naïve patients, with resistance rates of 10% for clarithromycin and 23% for metronidazole [10]. In Portugal, where one study estimated the overall clarithromycin resistance to be 50%, higher rates were associated with failed eradication treatment [11].

North American studies on the antibiotic resistance rates for *H. pylori* remain scarce [2]. A recent meta-analysis restricted to US-only studies (n = 19 studies total) found high resistance to clarithromycin at 31.5% (95% CI 23.6%-40.6%; *n* = 18 studies) and metronidazole at 42.1% (95% CI 27.3%–58.6%; *n* = 14 studies) [12]. However, there was considerable heterogeneity across the studies used to estimate the pooled prevalence rates in these meta-analyses. As with a 2018 meta-analysis across the Americas, the resistance prevalence rates were lower when restricted to treatment-naïve patients: 16.7% for clarithromycin and 29.3% for metronidazole [12]. More work is necessary to understand the drivers of the variable resistance rates across the US.

Even less is known about resistance rates among rural and Indigenous populations. In an *H. pylori* resistance surveillance study among Alaska Natives, metronidazole resistance was observed in 42% of cultured isolates, followed by clarithromycin resistance in 29.8% of isolates [13]. None of these prior studies included data from American Indian populations living in the lower 48 states. In this current cross-sectional study, we utilized biopsy samples from patients already undergoing an upper gastrointestinal endoscopy at a clinic serving American Indians in northern Arizona where *H. pylori* prevalence is near 60% [14] and about 30% of gastric biopsy samples were *H. pylori*-positive [15]. Navajos in the American Southwest are 3.5 times at higher risk of developing gastric cancer than non-White Hispanics [14]. Identifying antibiotic resistance in *H. pylori* could lead to a screen-and-treat strategy in these high-risk communities and to improved therapies. Microbial DNA from gastric biopsies was amplified to assess mutation rates for clarithromycin and metronidazole associated with *H. pylori* antibiotic resistance. The prevalence of mutations associated with infection was compared by age and sex for each antibiotic. Our goal was to provide an insight into antibiotic resistance in this highly at-risk population and the observed high rate of treatment failure. This information can be useful for assessing treatment choices.

## 2. Materials and Methods

### 2.1. Ethical Considerations

Working with medical samples and with American Indian communities requires IRB oversight and approval from the communities within which we work. The protocol and consent documentation were first approved by the Northern Arizona University Institutional Review Board (IRB #8195001). We also received chapter resolutions before seeking Navajo Nation Human Research Review Board (NNHRRB) protocol approval (NNR-16.263 approved 21 August 2021). Approval by the NNHRRB to submit the manuscript was received on 1 July 2023. All biopsy samples and data were deidentified prior to transport to Northern Arizona University where the analyses took place.

### 2.2. Sample Collection

Participants were recruited from the Winslow Indian Health Care Clinic (WIHCC) general surgery clinic during their visit to discuss esophagogastroduodenoscopy (EGD) for upper GI symptoms. WIHCC is located in and serves eight-chapter communities in the southwestern Navajo Nation. During their medical visit, the *H. pylori* study was explained to the patient by the surgeon. Willing participants signed an informed consent form to have two additional biopsy samples collected and their medical records abstracted for histopathology findings, prior testing and treatment for *H. pylori*, and demographic characteristics. Inclusion criteria were that the individual be at least 18 years of age and a patient. Clinic staff completed medical record abstraction using a record abstraction form and individuals were given a participant identification number for the sample and the abstracted information to protect anonymity. During endoscopy, a biopsy from both the antrum and the fundus was collected, placed in 400 μL RNAlater, and transported to the NAU laboratory for bacterial DNA isolation. All biopsy samples and data were deidentified prior to transport to Northern Arizona University.

### 2.3. PCR Amplification

The antrum and fundal samples were extracted separately, and only positive samples were processed. For analysis, the 23S ribosomal RNA (23S) and the oxygen-insensitive NADPH nitroreductase (*rdxA*) regions were sequenced for clarithromycin and metronidazole resistance, respectively (Table 1). First, the *glmM*, which encodes for a phosphoglucosamine mutase, was used to confirm that samples were PCR-positive for *H. pylori*. All PCR mixtures consisted of 10 µL of BioRad iTaq Universal SYBR Green Supermix (2X); 200 nM of each forward and reverse primer and one microliter of *H. pylori* DNA (20–50 ng) were added to each reaction mixture, with a final volume of 40 µL. Antibiotic-resistance-associated genes were run in a BioRad CFX96 real-time PCR system. The PCR programs for amplification consisted of 5 min at 95 °C, followed by 35 cycles of 30 s at 94 °C, 30 s at 55–63 °C, 40 s at 72 °C, and a final incubation at 72 °C for 3 min. In some cases, PCR products (7 µL of each sample) were electrophoresed in 2% agarose gels stained with SYBR Green for 1 h at 100 V.

### 2.4. Primers

Custom primers were developed to detect the resistance-associated mutations (Table 1). Clarithromycin-resistant mutations were defined as the presence of A2142G, A2143G, or T2182C mutations of the 23S ribosomal RNA [19,20,21,22]. Mutations associated with metronidazole were defined as R16H, R16C, R16P, H25R, H53R, H53A, D59N, L62V, A68T, A68V, A68S, A68N, G98S, G163V, G163D, V204I, and A206T [23]. Real-time PCR was used to amplify these sequences using the primers listed in Table 1 above. Sequencing of the amplified *H. pylori* DNA was performed using Genewiz (Azenta Life Sciences, South Plainfield, NJ, USA). The analytic method was validated using a commercially available resistance testing kit, the Viasure *H. pylori*-Clarithromycin resistance kit (Jant Pharmaceutical Corporation, Encino, CA, USA). Testing with this kit showed 100% sensitivity and specificity for the kit’s target mutations (A2142G, A2143G; see Appendix A). There is no commercial kit available to determine metronidazole resistance as point mutations seem to change depending on the population and country.

### 2.5. Analysis

The analysis was restricted to individuals with no history of *H. pylori* infection recorded in their medical records, and thus, we assumed the record to represent their primary infection. In addition, only samples for which the region of the pathogen, 23S ribosomal RNA for clarithromycin and *rdxA* for metronidazole, was successfully amplified are included in the analysis. Analyses were performed in triplicate for each sample.

Sample concordance was calculated as the number of pairs that were concordant (including those with no mutations) from all successfully amplified antrum/fundal pairs. For each antibiotic, mutations associated with antibiotic resistance were compared by age (*t*-test) and sex (chi square). Treatment is qualitatively described for patients for whom we had treatment information.

### 2.6. Statistical Analysis

Statistical analyses were performed using StataBE v17 (StataCorp LLC, College Station, TX, USA).

## 3. Results

### 3.1. Recruitment and Inclusion

Individuals were recruited into the study during their EGD appointment. The project and the process were discussed. After consenting to the project, two additional biopsy samples were taken from both the antrum and fundus for 396 individuals and medical records were abstracted. Based on the information abstracted from the medical records, patients for whom a history of *H. pylori* infection (e.g., prior positive test results were recorded or history of *H. pylori* was listed in the record) was noted were excluded (N = 167). After removing participants from this study because their samples were negative for *H. pylori* (N = 158) or could not be tested (N = 3), there were 67 participants with *H. pylori*-positive biopsy samples. Restricting the analysis to those samples which were *H. pylori*-positive, where the sample successfully amplified, and where the patient’s record showed no prior history of *H. pylori* yielded 49 individuals (64 samples) for clarithromycin and 33 individuals (41 samples) for metronidazole (Figure 1).

### 3.2. Antibiotic Clarithromycin

Clarithromycin resistance, defined as the presence of either T2182C, A2142G, or A2143G mutations, was present in 38.8% (n = 19) of the participants for whom 23S ribosomal RNA was successfully amplified. T2182C (n = 18) was the most observed mutation, followed by A2142G (n = 1) and A2143G (n = 1). There were no statistically significant differences between those with and without clarithromycin-resistant mutations regarding mean age (*p* = 0.18) or sex (*p* = 0.51; Table 2). If the resistance definition was restricted to the more conservative A2142G or A2143G, the resistance prevalence was 4% (n = 2 of 49 individuals).

Among the thirteen individuals for whom both the antrum and fundal biopsy samples successfully amplified, nine had no clarithromycin-associated mutations and three had the same mutations in both samples. The one discordant pair exhibited the T2182C mutation only in the antrum sample. Sensitivity analysis was performed using the Viasure clarithromycin resistance kit which confirmed the presence of A2142G and A2143G. We provided data to illustrate the use of this kit to differentiate and confirm clarithromycin-resistant positive samples targeting the two known mutations from wild-type isolates. The mutation T2182C is not detected by the kit. This kit was able to identify 100% of our clarithromycin-positive samples with A2142G and A2143G mutations. Also, the kit was able to identify 100% of clarithromycin-positive samples with A2142G/A2143G with another mutation, T2182C. When the mutation T2182C was analyzed by itself, it was not recognized by the kit (Table 3).

### 3.3. Antibiotic Metronidazole

Of the mutations associated with metronidazole resistance, we observed R16H (n = 4 individuals), R16C (n = 2), H53R (n = 17), D59N (n = 31), L62V (n = 22), and G98S (n = 15). Among the 33 individuals with no history of *H. pylori* infection, where the *rdxA* was successfully amplified, 93.9% (n = 31) showed at least one of these mutations. There was no observed difference in mean age or sex differences; however, the *H. pylori* DNA in only two individuals did not express any of the mutations of interest (Table 2).

Five of the seven individuals with mutations known to be resistant to metronidazole who had both biopsy samples successfully amplified in the *rdxA* gene had identical mutations in both the antrum and fundus biopsy pairs. Of the two discordant samples, one exhibited metronidazole-associated mutations only in the antrum sample, while the other shared one matching mutation.

### 3.4. Antibiotic Treatment

Because we are interested in patient outcomes, we also looked at the treatment regimes for participants in this study. Among the 67 individuals positive for *H. pylori* who met the inclusion criteria for this study, the prescribed treatment was known for 26 individuals (Figure 2). Triple therapy consisting of amoxicillin, clarithromycin, and pantoprazole was the most prescribed treatment regimen (57.7% of known treatments). Bismuth with metronidazole, doxycycline, and omeprazole was the next most prescribed treatment (23.1%).

## 4. Discussion

To our knowledge, this is the first contemporary survey of *H. pylori* resistance among American Indians in the lower 48 states. In this sample of American Indians from northern Arizona undergoing EGD, we found high rates of mutations associated with antibiotic resistance (38.6% of patients exhibited mutations associated with clarithromycin and 95.6% had mutations associated with metronidazole). Our analysis indicates concerning rates of resistance although the clinical phenotype for some of these mutations needs to be further addressed using samples where full culture and sensitivity is possible. The high prevalence of *H. pylori* in the population and the high frequency of resistance-associated mutations suggest that incorporating susceptibility testing is important when determining optimal treatment regimens.

The presence of the T2182C clarithromycin resistance mutation in this population is high compared with other populations [16]. Agudo et al. [24] found this mutation in only 5.9% of 118 patients in Madrid while finding higher A-to-G 2142/3 point mutations at rates of 88.1% and 85.3%, respectively [17]. The T2182C mutation seems to be predominant in Asian countries where the infection is also high and it may indicate the long-term association between *H. pylori* and our population [16,18], and further investigation of this mutation is warranted. While the A-to-G mutation at the 2142 or 2143 position or the A-to-C mutation at the 2142 region of the 23S rRNA gene is well established, the clarithromycin resistance role of the T2182C mutation has been questioned [19]. Specifically, while the mutation may be observed among individuals with clarithromycin resistance, culturing yields mixed sensitive and resistant colonies [20]. Others have shown that the association between established mutations is highly variable across populations and with mixed results even in culturing [17,18]. This is why whole genome sequencing has been proposed to identify multiple antibiotic resistance markers [21] and improve therapy outcomes.

Our findings are closer to the 93.2% metronidazole resistance rates observed among a high-risk Colombian population, though none of their samples were resistant to clarithromycin (n = 59) based on in vitro experiments and PCR amplification [22]. Other studies in high-risk populations have found higher clarithromycin resistance, for example, 31.2% in a study in Santiago, Chile, where *H. pylori*-positive rates were 48.7% [23]. While studying the pattern of antibiotic resistance in Mexican mestizo for 20 years, it was found that while metronidazole resistance decreased from 75% to 51%, resistance for clarithromycin increased from 10% to 32% [25]. Our results have provided a limited insight into antibiotic resistance in northern Arizona, and it may help explain some of the reasons for treatment failure while emphasizing the need for further studies.

*H. pylori* eradication can prevent gastric cancer. However, the value of empirical standard triple therapy in this population is unknown because data on antibiotic resistance is lacking. Based on commonly used antibiotics used to treat this infection, two approaches that can improve eradication therapy are the identification of clarithromycin- or metronidazole-resistant *H. pylori* isolates and the implementation of tailored therapies where suspected resistance rates are higher for these antibiotics. Although these approaches would be ideal, culturing *H. pylori* is cumbersome, time-consuming, and rarely performed in clinical laboratories in the Navajo Nation. The lack of bacterial cultures rules out the implementation of antibiotic susceptibility tests. Whole genome sequencing (WGS) has also been proposed to identify resistance patterns in clinical *H. pylori* isolates [26], but WGS is expensive and not routinely performed in clinical laboratory settings.

Furthermore, surveillance data for *H. pylori* antibiotic resistance rates in this population and the USA are scarce. In the Navajo Nation, the prevalence of *H. pylori* is over 60% and needs to be addressed, as infection with *H. pylori* presents a high risk of gastric cancer. However, *H. pylori* infection alone cannot explain the progression to gastric cancer, and other risk factors in this population cannot be ignored [14]. The alternative is to consider all *H. pylori* harmful and treat them regardless of their antibiotic characterization. Surgeons at Indian Health Services have proposed this approach to diminish the disproportionately high rate of *H. pylori* infection and gastric cancer in the Navajo Nation. Studies are also needed to evaluate the efficacy of widespread eradication programs to improve gastric cancer outcomes.

*Limitations.* These results are representative of patients undergoing EGD at a general surgery clinic in northern Arizona. We restricted the analysis to those individuals who did not have a history of *H. pylori* infection in their medical records. While this limited the sample size, it provided an estimate of antibiotic resistance in treatment-naïve patients. It is possible that this reflects a higher use of macrolides during adolescence than in older patients. It would be interesting to compare resistance rates for those with primary or persistent infections.

Like most *H. pylori* antibiotic resistance studies [12], the resistance estimates were generated using molecular sequencing for the presence of resistance-associated genes rather than by culture, which is used to determine minimum inhibitory concentrations. This is a limitation as it measures the existence of resistance mutations in the *H. pylori* DNA rather than clinical evidence of resistance. Metronidazole has shown a general lack of reproducibility in in vitro testing. Neither Etest nor agar dilution is recommended as a reliable means for assessing *H. pylori* resistance to metronidazole [24]. In meta-analyses, stratifying by diagnostic method has not shown significant differences in the estimated resistance rates [9,10].

These findings regarding antibiotic resistance are clinically relevant to the choices around treatment for *H. pylori*. At the time of this study, we had treatment information for only 39% of the individuals who met the inclusion criteria. This low percentage may be because of the timing of record abstraction or may be indicative of differences in the sensitivity of clinical diagnostic tests. DNA extraction is more sensitive than both histopathology and the rapid CLO tests performed in clinics, which may mean patients go undetected and untreated using clinical diagnostics [27]. However, given the observed resistance mutations in this sample and that more than half of the treatments included clarithromycin (57.7% of known treatments) and another quarter (23.1%) of treatments included metronidazole is cause for pause. It should be noted that treatment recommendations do change. We did not look at temporal trends in the treatment regimes over the study period. Current guidelines recommend bismuth and non-bismuth quadruple concomitant (PPI, amoxicillin, clarithromycin and a nitroimidazole) as a first line of treatment in areas with clarithromycin resistance over 15% [8]. Others have reported that clarithromycin should be avoided in countries where resistance prevalence is greater than 25% [9]. Our findings highlight the need for improved understanding of antibiotic resistance profiles.

Depending on the definition of clarithromycin resistance, the observed resistance was either well above both guideline thresholds, that is, 38.6% when including T2182C in the definition of resistance, or well below them—4% when restricting to just A2142G or A2143G mutations. Perhaps a recommended way to address this issue is by reporting the association of mutations at A2143G and at A2142G with the T2182C mutation [28]. A recent report from Asia investigating point mutations in the 23S rRNA gene and failure of clarithromycin treatments has proposed the use of the A2143G—T2182C combination as a predictor phenotype of clarithromycin eradication failure [22]. When this combination was analyzed, treatment failure increased from 20.9% to 57.7%. Furthermore, it was also reported that the presence of T2182C without A2143G reduced the failure rate to 4%. Because this point mutation is widely found in our study population, a larger study is needed to determine if this mutation could be used as a predictor for success or failure in clarithromycin-based eradication regimens.

While a screen-and-treat strategy is clearly recommended in high-risk communities, the implication of antibiotic resistance means that a screen-and-treat policy for communities with an intermediate–low risk of gastric cancer is only weakly supported [8]. A Norwegian cohort showed an increased frequency of metronidazole (69.5%) and clarithromycin (38.5%) resistance among those who failed treatment [29]. As susceptibility testing becomes increasingly available, it can guide initial therapies where empiric therapy success rates are low and should be considered for patients with a history of prior treatment failure [30].

## 5. Conclusions

Given the risk of treatment failure with antibiotic resistance, there is a need to consider the resistance profile of the patient population during treatment. Little is known about resistance rates among rural indigenous populations in northern Arizona, where *H. pylori* prevalence is near 60% and about one-third of gastric biopsy samples are *H. pylori*-positive. The resistance rates we observed in this population of patients undergoing endoscopy are similar to those reported in other high-risk populations: 38.8% for clarithromycin and 93.9% for metronidazole. Given the high prevalence of T2182C in our study population, further studies would be needed to determine its functional role in clarithromycin-based treatment regimens.

## Figures and Tables

**Figure 1 microorganisms-11-02290-f001:**
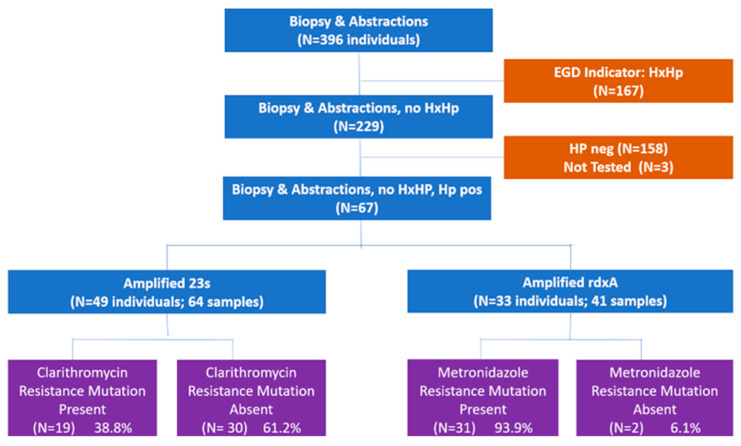
Participant recruitment and resistance results. (HxHp—history of *H. pylori*; neg—negative, pos—positive).

**Figure 2 microorganisms-11-02290-f002:**
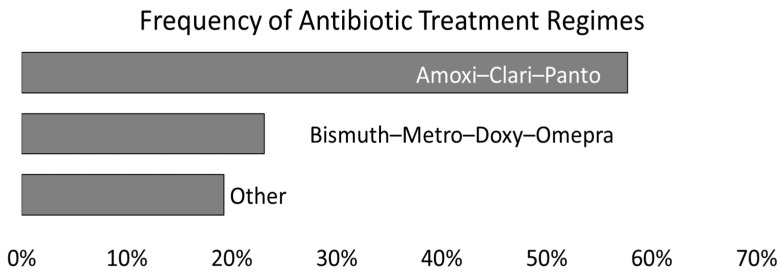
*H. pylori* treatment regime for participants meeting the inclusion criteria for whom treatment information was available (n = 26).

**Table 1 microorganisms-11-02290-t001:** Primers used for the amplification of *H. pylori*, as well as those genes involved in clarithromycin·(235)·and-metronidazole (*rdx*A) resistance.

Target	Gene	Nucleotide Sequence	Product Size·(bp)	Ref.
*H. pylori*	*glmM*	F–5′-AAGCTTTTAGGGGTGTTAGGGGTTT-3′ R–5′-AAGCTTACTTTCTAACACTAACGC-3′	300 bp	[16]
Clarithromycin	23S	F–5′-CCACAGCGATGTGGTCTCAG-3′ R–5′·CTCCATAAGAGCCAAAGCCC-3′	425 bp	[17]
Metronidazoler	*rdxA*	F–5′-GCAGGAGCATCAGATAGTTCT-3′ R–5′-GGGATTTTATTGTATGCTACAA-3′	886 bp	[18]

**Table 2 microorganisms-11-02290-t002:** Known resistant mutations in region 23S and *rdxA*. This analysis is restricted to those for whom the *H. pylori* DNA was successfully amplified.

**Clarithromycin** **(23S)**	**All** **(n = 48)**	**Mutation** **(n = 19)**	**No Mutation** **(n = 29)**	
Age, mean (sd)	54.2 (14.6 sd)	50.8 (15.7 sd)	56.6 (13.6 sd)	t = 1.4 *p* = 0.178
Sex				
F	23 (47.9%)	11 (57.9%)	15 (51.7%)	*X*^2^ = 0.00
M	25 (52.1%)	8 (42.1%)	14 (48.3%)	*p* = 0.51
**Metronidazole** **(*rdxA*)**	**All** **(n = 33)**	**Mutation** **(n = 31)**	**No Mutation** **(n = 2)**	
Age mean (sd)	53.3 (15.6 sd)	53.0 (16.0sd)	57.5 (2.1 sd)	t = 0.4 *p* = 0.70
Sex				
F	16 (48.5%)	15 (48.4%)	1 (50%)	*X*^2^ = 0.00
M	17 (51.5%)	16 (51.6%)	1 (50%)	*p* = 0.965

**Table 3 microorganisms-11-02290-t003:** Testing of clarithromycin-resistant *H. pylori using* the commercial kit from Viasure that measures resistance to clarithromycin by targeting two known mutations, A2142G and A2143G.

		Viasure Real Time PCR Kit	
Mutations	n	Positive	Negative
T2182C	28	0	100%
A2142G	5	100%	0%
A2143G	4	100%	0%
A2142G + T2182C	1	100%	0%
A2143G + T2182C	4	100%	0%
10 Hp (+) samples	10	100% (6/10)	(4/10)
		(Confirmed by sequencing)	
Hp (−)	40	0%	100%

## Data Availability

The datasets generated and/or analyzed during the current study are available from the corresponding author upon reasonable request.

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
