# Peer review of "Antibiotic Resistance Rates for Helicobacter pylori in Rural Arizona: A Molecular-Based Study"

_microorganisms, 2023, doi:10.3390/microorganisms11092290_

Round 1
Reviewer 1 Report
The authors reported the proportion of Helicobacter pylori antibiotic-resistant bacteria among American Indians in Arizona. The presence or absence of resistant bacteria is very important in determining eradication treatment regimens. The small sample size is an issue, but this may be unavoidable due to the rarity of the subject. No further modifications are considered necessary. Additional discussion on the reasons for the emergence of antibiotic-resistant would be better.
Author Response
Thank you very much for your kind suggestion. Indeed, your comment about presence or absence dictating eradication treatment is true. Unfortunately, little has been done by different healthcare centers in the Navajo Nation to address this issue. We hope this manuscript will bring awareness about new studies needed that can drive proper eradication that could explain the persistent infections seen in this patient population.
Reviewer 2 Report
Dear authors,
Your paper entitled "Antibiotic resistance rates for Helicobacter pylori in rural Arizona: a molecular-based study" add a new insight into antibiotic resistance in Helicobacter pylori in the studied population. As you said, a further study in which is correlated genotypic resistance with phenotypic one will be more than needed, but even so, your study add information to the present knowledge regarding antimicrobial susceptibility in this bacterium.
It is a small study but rise a question about the importance of mutation T2182C in clarithromycin resistance.
One small error: line 115 -" ...using the primers listed below." Actually, the primers used are presented in Table 1 which is above this line (107-108).
Author Response
It is a small study but rise a question about the importance of mutation T2182C in clarithromycin resistance.
The role of the T2182C is novel in this population but has been addressed in Asian countries as a potential indicator of clarithromycin treatment failure when combined with mutations at 2142 position. I would love to see this interaction in the Alaska Native population where antibiotic-resistant rates are much higher than those observed in the Navajo Nation.
One small error: line 115 -" ...using the primers listed below." Actually, the primers used are presented in Table 1 which is above this line (107-108).
Thank you. This has been corrected in the manuscript.